# Expression of *CYP2B6* Enzyme in Human Liver Tissue of HIV and HCV Patients

**DOI:** 10.3390/medicina59071207

**Published:** 2023-06-27

**Authors:** Bozana Obradovic, Owain Roberts, Andrew Owen, Ivana Milosevic, Natasa Milic, Jovan Ranin, Gordana Dragovic

**Affiliations:** 1University of Belgrade, Faculty of Medicine, Department of Pharmacology, Clinical Pharmacology and Toxicology, 11000 Belgrade, Serbia; gozza@beotel.net; 2University of Buckingham Medical School, Faculty of Medicine and Health Sciences, University of Buckingham, Buckingham MK18 1EG, UK; owain.roberts@buckingham.ac.uk; 3Centre of Excellence in Long-Acting Therapeutics (CELT), Department of Pharmacology and Therapeutics, Institute of Systems, Molecular and Integrative Biology, University of Liverpool, Liverpool L69 3BX, UK; aowen@liverpool.ac.uk; 4Faculty of Medicine, University of Belgrade, 11000 Belgrade, Serbia; ivana.milosevic00@gmail.com (I.M.); silly_stat@yahoo.com (J.R.); 5Clinic of Infectious and Tropical Diseases, Clinical Centre of Serbia, 11000 Belgrade, Serbia; 6University of Belgrade, Faculty of Medicine, Department of Medical Statistics & Informatics, 11000 Belgrade, Serbia; njranin@gmail.com

**Keywords:** HCV, HIV, HCV/HIV co-infection, metabolic enzymes, metabolic transporters, *CYP2B6*, *CYP3A4*, *ABCB1*, DAA, gene expression

## Abstract

*Background and Objectives*: Hepatitis C virus (HCV) and human immunodeficiency virus (HIV) infections present significant public health challenges worldwide. The management of these infections is complicated by the need for antiviral and antiretroviral therapies, which are influenced by drug metabolism mediated by metabolic enzymes and transporters. This study focuses on the gene expression of *CYP2B6*, *CYP3A4*, and *ABCB1* transporters in patients with HIV, HCV, and HIV/HCV co-infection, aiming to assess their potential association with the choice of therapy, patohistological and clinical parameters of liver damage such as the stage of liver fibrosis, serum levels of ALT and AST, as well as the grade of liver inflammation and other available biochemical parameters. *Materials and Methods*: The study included 54 patients who underwent liver biopsy, divided into HIV-infected, HCV-infected, and co-infected groups. The mRNA levels of *CYP2B6*, *CYP3A4*, and *ABCB1* was quantified and compared between the groups, along with the analysis of liver fibrosis and inflammation levels. *Results*: The results indicated a significant increase in *CYP2B6* mRNA levels in co-infected patients, a significant association with the presence of HIV infection with an increase in *CYP3A4* mRNA levels. A trend towards downregulation of *ABCB1* expression was observed in patients using lamivudine. *Conclusions*: This study provides insight into gene expression of *CYP2B6 CYP3A4*, and *ABCB1* in HIV, HCV, and HIV/HCV co-infected patients. The absence of correlation with liver damage, inflammation, and specific treatment interventions emphasises the need for additional research to elucidate the complex interplay between gene expression, viral co-infection, liver pathology, and therapeutic responses in these particular patients population.

## 1. Introduction

Hepatitis C virus (HCV) infections and human immunodeficiency virus (HIV) infections pose substantial challenges to global public health, affecting millions of individuals worldwide. These infections not only lead to significant morbidity and mortality but also complicate the management of coexisting medical conditions, including the need for antiretroviral therapy in HIV-infected patients and antiviral therapy in HCV-infected patients [1]. HIV infection is characterized by severe immunological dysfunction and unbalanced T-cell homeostasis [2]. Since the development of effective combination antiretroviral therapy (ART), life expectancy has equaled that of persons who are not infected with the virus. Early treatment has resulted in rapid and effective viral suppression, increased treatment adherence and persistence, longer life expectancy, a decrease in the occurrence of HIV-related co-morbidities reduction of complications; and probably most importantly, a decrease in the spread of HIV [2,3]. In the past five years, there has been substantial progress made in the management of HCV. However, patients co-infected with HCV and HIV face a more complex clinical scenario compared to those with either infection alone [4]. Co-infection with HIV and HCV has complex implications for disease progression, treatment outcomes, and overall patient management. The shared routes of transmission contribute to the increased likelihood of co-infection in these populations [1,4]. The pathogenesis of HIV/HCV co-infection is complex and multifaceted. HIV-induced immunosuppression plays a crucial role in HCV replication and disease progression. Immune dysregulation caused by HIV leads to impaired control of HCV infection, resulting in higher HCV viral loads and accelerated liver disease [5]. Conversely, HCV infection also impacts HIV disease progression by influencing viral replication and immune response [6]. The presence of HIV accelerates the progression of HCV-related liver disease, leading to an increased risk of cirrhosis, hepatocellular carcinoma, and liver-related mortality [4,6].

The efficacy and safety of therapeutic interventions in these populations are profoundly influenced by drug metabolism, which is governed, in part, by metabolic enzymes and transporters. Cytochrome P450 2B6 (*CYP2B6*) is a hepatic enzyme involved in the biotransformation of numerous drugs, including antiretroviral and antiviral agents such as efavirenz, tenofovir/adefovir and lamivudine [7]. Moreover, certain HCV direct-acting antivirals (DAA), including velpatasvir and dasabuvir, are also metabolized by *CYP2B6* [8]. Cytochrome P450 3A4 (*CYP3A4*), a major enzyme responsible for the metabolism of various drugs, also metabolizes numerous antiretrovirals [9,10], as well as several HCV DAAs, including grazoprevir and voxilaprevir [11]. The ATP-binding cassette subfamily B member 1 (*ABCB1*) transporter, commonly known as P-glycoprotein (P-gp), plays a critical role in drug absorption and the elimination of a variety of drugs including several antiretrovirals and antivirals, thereby influencing their bioavailability.

Both HIV and HCV infections can modulate *CYP2B6*, *CYP3A4*, and *ABCB1* gene expression levels through complex mechanisms involving viral proteins, inflammatory mediators, and cytokines. Consequently, the altered expression and activity of *CYP2B6*, *CYP3A4*, and *ABCB1* in HIV and HCV-infected individuals can significantly impact drug pharmacokinetics and treatment outcomes. The available in vitro and animal model studies have shown the influence of inflammation on the activity of CYP P450 enzymes, through multiple and complex transcriptional and post-transcriptional mechanisms [11,12]. Data from the literature primarily indicate the inhibitory effect of inflammation on the activity of enzymes [12,13], but some studies have shown the induction of liver enzyme activity in the presence of a higher degree of inflammation [14]. The nature of inflammatory mediators, i.e., the origin of inflammation and duration of inflammation, can also contribute to the changed activity of metabolic enzymes and transporters [12,13].

Limited research has been conducted to evaluate the gene expression of metabolic enzymes and transporters in patients with HIV, HCV, and HCV/HIV co-infection, and to best to our knowledge, very few were conducted on human liver tissue. Previous studies have found that gene expression levels, as an indicator of enzyme activity, exhibit less variability compared to protein abundance, enzyme catalytic activity, or the presence of polymorphisms [15,16,17,18]. Comparing the expression of genes encoding metabolic enzymes and transporters with constitutive (housekeeping) genes can provide insights into their potential patterns of expression in this specific population. Therefore, the aim of this study was to analyze the gene expression of *CYP3A4*, *CYP2B6*, and *ABCB1* transporters in a group of HIV, HCV, and HIV/HCV co-infected patients on human liver samples. An additional aim was to analyze the association between the degree of liver fibrosis, level of liver inflammation, and potential impact of applied antiviral and antiretroviral therapeutics on the expression of *CYP3A4*, *CYP2B6*, and *ABCB1* transporters in this patient population.

## 2. Materials and Methods

### 2.1. Demographic and Clinical Data

This cross-sectional study included 54 patients who underwent liver biopsy at the Clinic for Infectious and Tropical Diseases, “Dr. Kosta Todorovic”, University Clinical Center of Serbia, Belgrade, Serbia. The patients were divided into the following three groups: HIV-infected patients, HCV-infected patients, and patients co-infected with HIV and HCV. All HIV patients, both co-infected and mono-infected, received one of the following cART combination: two nucleoside/nucleotide reverse transcriptase inhibitors (NRTIs) (lamivudine + abacavir) + one drug from non-nucleoside/nucleotide reverse transcriptase inhibitors (NNRTIs) (efavirenz) or two NIRTs (emtricitabine + tenofovir) + one integrase inhibitor (INSTI) (dolutegravir). Co-infected patients did not receive anti-HCV treatment and HCV mono-infected patients were divided into subgroups of treated and untreated patients. HCV treated patients received pegylated interferon (PEG IFN) and DAA, (sofosbuvir/velpatasvir). HIV/HCV co-infected patients were compared with HCV mono-infected patients who did not receive HCV therapy and with a group of HIV mono-infected patients (receiving cART). In the HCV mono-infected group of patients, a comparison was performed between patients who received treatment (PEG IFN and DAA) and patients who did not receive treatment. A comparative analysis was conducted among these groups on various levels. Messenger RNA (mRNA) levels of *CYP2B6*, *CYP3A4*, and *ABCB1* were assessed and compared between the groups. Furthermore, all the patient groups were analyzed for liver fibrosis and liver inflammation level and then compared with mRNA levels in between groups. Additionally, other parameters available for analysis included information on cART and anti-HCV treatment, CD4+ T-cell count, platelet count, levels of liver enzymes (aspartate aminotransferase, AST, and alanine transaminase ALT), total bilirubin, total cholesterol, triglycerides, low-density lipoprotein (LDL), high-density lipoprotein (HDL), C-reactive protein (CRP) values, body mass index (BMI), alcohol and smoking consumption. As for the additional medication used in this study population, one patient from a group of HCV untreated patients was taking Lagosa^®^ (Silymarin, 420 mg q.d.), two patients from a group of HCV treated patients were taking lorazepam (1 mg q.d.) and aspirin (100 mg q.d.), respectively, one patient in the group of co-infected HIV/HCV was taking Lagosa^®^ (Silymarin, 420 mg q.d.), and one patient from a group of HIV mono-infected patients was taking aspirin (100 mg q.d.) and bisoprolol (1.25 mg q.d.).

### 2.2. Inclusion and Exclusion Criteria

The study’s patient selection process involved selecting individuals from a pre-established group who had previously undergone screening and met the requirements for liver biopsy. Inclusion criteria were age older than 18, male and female and serologically confirmed diagnosis of HIV and HCV infection. Additionally, PCR was used for confirmation of diagnosis. Exclusion criteria were hepatitis of unexplained etiology, autoimmune hepatitis, associated liver diseases, and use of all known drugs or substances that are inhibitors or inducers of liver enzymes (*CYP2B6* and *CYP3A4*) and transporters (*ABCB1*). All patients signed informed consent for participation in the study in accordance with the guidelines included in the Declaration of Helsinki and with approval from the Ethics Committee of the Faculty of Medicine, University of Belgrade (decision no. 17/I-8).

### 2.3. Liver Biopsy

A liver biopsy was performed as part of the diagnostic and therapeutic protocols in patients for whom it was indicated. The purpose of the liver biopsy was to evaluate the level of necroinflammatory activity (inflammation grading) and determine the extent of liver damage (fibrosis) in patients who had already been diagnosed with HIV, HCV, or HIV/HCV co-infection. Biopsy was performed using a Menghini 14-gauge needle. Each tissue cylinder obtained by biopsy was at least 2 mm wide and at least 20 mm in length. After sampling, the liver tissue was divided into two parts. One part was fixed in formalin and used to determine the stage of liver fibrosis and the grade of liver inflammation. The remaining liver tissue was snap-frozen over liquid nitrogen and stored at a temperature of −80 °C until qRT-PCR analyses were performed. All samples were collected at the VI and XI departments at the Clinic for Infectious and Tropical Disease “Dr. Kosta Todorovic”, UCCS, Belgrade, Serbia, and stored at the Department of Pharmacology, Clinical Pharmacology and Toxicology, Medical Faculty, University of Belgrade, Belgrade, Serbia.

### 2.4. Tissue Homogenization and RNA Extraction from Liver Tissue

All procedures were carried out in an RNase-free environment and all solutions were made up using Rnase-free water/reagents. QIAzol Lysis Reagent (Qiagen, Life Technologies, Manchester, UK) was used for homogenization on Minilys^®^ personal homogenizer (Bertin Instruments, Rockville, MD, USA), as well as chloroform, isopropanol, and 75% ethanol. The RNA pellet was dissolved in the appropriate volume of Rnase-free water. For each sample, the RNA Clean-Up Protocol was performed according to manufacturer instructions.

### 2.5. cDNA Synthesis

For cDNA synthesis, TaqMan^®^ Reverse Transcription Reagents (Thermofisher, Cat No. N8080234, Inchinnan, Scotland, UK) were used. RNA quality and quantity were assessed by spectrophotometry using a NanoDrop ND-1000 spectrophotometer (Thermo Scientific, Waltham, MA, USA).

### 2.6. Gene Expression—Real-Time qRT-PCR and Data Analysis

For each condition to be tested, cDNA + H_2_O in combination were pipetted into two separate sterile, nuclease-free 0.2 mL PCR tubes (one was used for the test gene and one was used for the housekeeping gene) and mixed well by vortexing. qPCR Mastermix was prepared for each gene to be studied, e.g., glyceraldehydes-3-phosphate dehydrogenase GAPDH (housekeeping gene) and the test gene (*CYP3A4*, *CYP2B6*, and *ABCB1*), and added to each sample tube. Expression via RT-qPCR was analyzed for all genes on 96-well white Hard-Shell™ PCR plates (Bio-Rad, Hemel Hempstead, UK, Cat. No. HSP-9601) using Mastermix (TaqMan^®^ Gene Expression Master Mix; Applied Biosystems, Waltham, MA, USA). All samples were run in triplicate. The following TaqMan^®^ Gene Expression Assay (Life Technologies) probes were used: *CYP2B6*: Hs04183483_g1; *CYP3A4*: Hs00604506_m1; *ABCB1*: Hs00184500_m1, and (GAPDH): Hs02758991_g1. Quantification was performed using Bio-Rad Opticon Monitor™ Analysis software (version 3.1.32). Fold changes in gene expression were determined as described by Pfaffl [19]. Change above 2 fold and below 0.5 fold was considered increased or decreased gene expression, respectively. Tissue homogenisation and RNA extraction from liver tissue, cDNA synthesis, and real-time qRT-PCR were performed at the Department of Translational Medicine at the University of Liverpool.

### 2.7. Histopathological Processing

After being fixed in formalin using standard pathohistological procedures, the tissues were embedded in paraffin blocks. They were then subjected to staining using the conventional hematoxylin–eosin method for subsequent pathohistological analysis. To assess the degree of fibrosis, sections from each block were stained by impregnating reticulin fibers with silver. Subsequently, contrast staining with hematoxylin was applied. The main histological parameters for assessing liver damage in biopsy material were the stage of fibrosis and the degree of liver inflammation. The level of liver fibrosis was classified based on the METAVIR score, as the absence of fibrosis, the presence of mild, moderate, severe fibrosis, and cirrhosis (stages F0–F4). The grade of inflammation was classified based on the METAVIR score as the absence of inflammation, the presence of minimal changes, and light, moderate, and high-intensity inflammation. Data on liver fibrosis and liver inflammation levels were obtained from the patient’s records. Representative liver tissue photomicrographs are presented in Figure 1.

### 2.8. Statistical Analysis

Categorical variables were presented as absolute numbers with percentages. Numeric variables were presented as means with standard deviations or medians with 25th–75th percentile according to the data distribution. Differences in numerical variables between the two groups were assessed by Student’s *t*-test or a Mann–Whitney test, as appropriate, while ANOVA with Least Significant Difference (LSD) as a posthoc test was used for assessing differences between three groups of patients. Categorical data were analyzed using a Chi-square test and Fisher exact test. Correlations were examined by correlation coefficients (Pearson or Spearman correlation coefficient) according to the data scale used in the analyses. The level of significance was set at 0.05. Statistical analysis was performed using the IBM SPSS 21 (Chicago, IL, USA, 2012) package. Data analysis was performed at the Department of Medical Statistics & Informatics, Medical Faculty, University of Belgrade, Serbia. Additional information on the methodology is available in Appendix A.

## 3. Results

### 3.1. Study Population

This cross-sectional study included a total of 54 patients in the following three groups: mono-infected HIV, mono-infected HCV, and HIV/HCV co-infected patients. The sample included 30 (55.6%) HCV-infected patients, 9 (16.7%) patients infected with HIV, and 15 (27.8%) patients co-infected with HIV/HCV. In the group of HCV patients, 14 (46.7%) received therapy and 16 (53.3%) were untreated. Among the treated HCV patients, seven (50.0%) received PEG IFN and seven (50.0%) received DAA. All HIV patients, mono-infected as well as patients with HCV co-infection, received antiretroviral therapy (ART). In total, 12 (50%) patients were treated with lamivudine, 11 (45.8%) with dolutegravir, and 14 (58.3%) with efavirenz, used as part of cART. None of the co-infected patients received treatment for HCV. The average patient age was 50 ± 13, while 34 (63.0%) were male, and 20 (37.0%) were female. All patients were white, with a median BMI of 23.6 (25th–75th percentile: 20.4–25.8) kg/m^2^. The median CD4+ T-cells count was 656 (25th–75th percentile: 459–928) cells/ml, median ALT of 96 (25th–75th percentile: 69–120) U/L; median AST of 60.5 (25th–75th percentile: 38–89) U/L. In total, 22 (40.7%) patients were smokers, while 12 (22.2%) reported alcohol consumption. The summary of the main clinical data of the study population is available in Appendix A and data on antiretroviral, anti-HCV and other medications for all patients are available in Appendix A.

### 3.2. CYP2B6 Expression in the Study Population

#### 3.2.1. Expression of *CYP2B6* in Groups of HIV, HCV, and HIV/HCV Co-Infected Patients

Less than a 2-fold increase in expression for *CYP2B6* was observed in 64.3% of untreated HCV patients, 51.7% of HIV patients, and 18.2% of HIV-HCV co-infected patients. The initial variance analysis indicated a significant difference in gene expression levels for *CYP2B6* (*p* = 0.017) between all three groups of patients, HIV, HCV, and HIV/HCV co-infected. A further detailed investigation has demonstrated a significant (*p* = 0.007) difference in *CYP2B6* expression in-between groups of untreated HCV mono-infected patients and HIV/HCV co-infected patients, suggesting increased expression in a group of co-infected patients (Table 1). An over 2-fold increase in *CYP2B6* expression was observed in 81.8% of HIV/HCV co-infected patients.

Similarly, our results have shown a significant difference (*p* = 0.030) in *CYP2B6* expression between groups of HIV/HCV co-infected patients and HIV mono-infected patients, showing increased expression in a group of co-infected patients. The level of *CYP2B6* expression showed no significant correlation (*p* > 0.05) to the grade of inflammation and stage of liver fibrosis, nor a significant difference was observed when compared between the groups of patients.

#### 3.2.2. *CYP2B6* Gene Expression in the Groups of HCV-Infected Treated and Untreated Patients

In the group of HCV-infected patients, ≥2-fold increase in expression for *CYP2B6* was observed in 35.7% of untreated HCV patients, while in the group of HCV treated patients, ≥-2-fold increase in *CYP2B6* was found in 42% of patients.

The results of this study have shown no significant difference (*p* = 0.252) in the expression of genes encoding *CYP2B6* in a comparison of groups of treated and untreated HCV mono-infected patients (Table 1). In addition, there was no significant difference (*p* > 0.05) in the grade of inflammation and stage of liver fibrosis nor in the values of biochemical and clinical parameters between these two groups.

#### 3.2.3. Independent Correlations of *CYP2B6*

In addition to these findings, the presence of HIV/HCV co-infection independently correlated (*p* < 0.05) with the increased expression of genes encoding *CYP2B6*.

Amongst the group of treated HCV mono-infected patients, univariate analysis has shown that the choice of therapy independently correlated with levels of *CYP2B6*. Namely, higher expression of *CYP2B6* genes was observed in a group of patients treated with PEG IFN compared to the group of untreated HCV-infected patients (*p* = 0.021), and in comparison with the group of HCV-infected patients treated with DAA (*p* = 0.028) (Table 1).

The level of *CYP2B6* expression showed no significant correlation (*p* > 0.05) to the grade of inflammation and stage of fibrosis of liver biopsies, nor was a significant difference observed when compared between the groups of patients. There was no significant correlation (*p* > 0.05) of *CYP2B6* found with values of biochemical parameters (thrombocyte and CD4+ lymphocyte count, levels of ALT and AST) nor with parameters such as BMI values, alcohol and tobacco consumption.

### 3.3. CYP3A4 Gene Expression in the Study Population

#### 3.3.1. *CYP3A4* Gene Expression in Groups of HIV, HCV and HIV/HCV Co-Infected Patients

Less than a 2-fold increase in expression for *CYP3A4* was observed in 38.5% of HCV patients without HCV therapy, while an over 2-fold expression increase was found in 71.4% of HIV-infected and 81.8% of HIV/HCV co-infected patients. Analysis has not shown a significant difference (*p* = 0.359) in *CYP3A4* expression between groups of HCV, HIV, and HCV/HCV+ co-infected patients (Table 2). The level of *CYP3A4* expression showed no significant correlation (*p* > 0.05) to the grade of inflammation and stage of liver fibrosis, nor was a significant difference observed when compared between the groups of patients.

#### 3.3.2. Expression of *CYP3A4* Gene in between Groups of HCV Treated and Untreated Patients

In the HCV group, an over 2-fold increase in expression for *CYP3A4* was observed in 61.5% of untreated HCV patients, and in 33.3% of treated HCV patients. The results of this study have shown no significant difference (*p* = 0.149) in the expression of genes encoding *CYP3A4* in a comparison of groups of treated and untreated HCV patients (Table 2). The level of *CYP3A4* expression showed no significant correlation (*p* > 0.05) to the grade of inflammation and stage of fibrosis of liver biopsies, nor was a significant difference observed when compared between the groups of patients. There was no significant correlation (*p* > 0.05) of CYP3A gene expression found with values of biochemical parameters (thrombocyte and CD4+ T-cell count, levels of ALT and AST) nor with parameters such asBMI values, alcohol and tobacco consumption.

#### 3.3.3. Independent Correlations of *CYP3A4* Gene Expression

Using univariant analysis, we confirmed the correlation of higher *CYP3A4* expression (*p* < 0.05) with the presence of HIV infection, including both groups of co-infected and mono-infected patients. Increased *CYP3A4* expression was in positive correlation (*p* < 0.05) with higher grades of liver inflammation, although no correlation (*p* > 0.05) with the stage of liver fibrosis was found. As for other potential factors affecting *CYP3A4*, platelet count was found to be significantly higher in patients with a higher expression of *CYP3A4*. There was no significant correlation (*p* > 0.05) between the changes in the level of expression of *CYP3A4* when compared with the other biochemical and clinical parameters analyzed.

### 3.4. ABCB1 Expression in the Study Population

#### 3.4.1. *ABCB1* Expression in Groups of HIV, HCV and HIV/HCV Co-Infected Patients

Less than a 2-fold increase in expression for *ABCB1* was observed in all cases of untreated HCV and HIV-infected patients, while ≥2-fold expression increase was found in 25% of HIV/HCV co-infected patients. The analysis did not show a significant difference (*p* > 0.05) in *ABCB1* expression between the groups of HIV, HCV, and HIV/HCV co-infected patients (Table 3). The level of *CYP2B6* expression showed no significant correlation (*p* > 0.05) to the grade of inflammation and stage of liver fibrosis, nor was a significant difference observed when compared between the groups of patients.

#### 3.4.2. *ABCB1* Expression in the Group of HCV Treated and Untreated Patients

In the group of HCV patients, none showed an over 2-fold increase in expression for *ABCB1*, for either the groups of treated and untreated patients HCV patients. The results of this study have shown no significant difference (*p* = 0.466) in the expression of genes encoding *ABCB1* in a comparison of groups of treated and untreated HCV patients (Table 3). Furthermore, there was no significant difference (*p* > 0.05) in the grade of inflammation and level of fibrosis, nor in the values of biochemical and clinical parameters between these two groups.

#### 3.4.3. Independent Correlations of *ABCB1* Expression

Borderline significance (*p* = 0.068) was observed, showing lower levels of expression of genes encoding *ABCB1* in patients treated with lamivudine (5.030 ± 6.580) in comparison to the levels of expression in patients who were not treated with this drug (1.039 ± 0.361). As for the other ART used, there was no significant correlation to the expression of *ABCB1* (*p* > 0.05).

Although a trend was observed when comparing differences in gene expression between groups of HCV patients and HIV/HCV co-infected patients, there was no significant correlation (*p* > 0.05) found. There was also no significant correlation (*p* > 0.05) found between other groups of patients compared in this study or when analyzed for correlation with the stage of liver fibrosis and grade of liver inflammation. Similarly, there was no correlation (*p* > 0.05) found when gene expression for the *ABCB1* transporter was compared with the other parameters that we followed.

## 4. Discussion

In this research, we conducted an analysis of human liver samples analyzing the expression of *CYP2B6*, *CYP3A4*, and *ABCB1* within groups of HIV, HCV, and HIV/HCV co-infected patients.

### 4.1. HIV/HCV Co-Infection Correlated with Increased Expression of CYP2B6

*CYP2B6* is primarily expressed in the liver tissue, showing significant interindividual variability and differences in activity depending on age, gender, race, as well as alcohol and smoking consumption [20]. In previous decades, studies have shown that *CYP2B6* accounts for 2–10% of the total hepatic CYP content and is involved in the metabolism of numerous known substrates, including antiretroviral drugs, such as efavirenz [21]. Of particular interest was the observation of increased *CYP2B6* expression in co-infected patients, which appeared to be independent of other potential factors. These findings are in accordance with the results of Pereira et al. who showed lower plasma concentrations of efavirenz in HIV/HCV co-infected patients compared with HIV mono-infected, therefore suggesting lower *CYP2B6* activity [22]. Notably, this heightened gene expression was not influenced by the use or choice of cART (Table 1). However, in patients with mono-infections of either HIV or HCV, there was no increase in *CYP2B6* gene expression (Table 1). Efavirenz plasma concentration has often been used as a parameter to determine *CYP2B6* metabolic capacity, as approximately 90% of this drug is metabolized by *CYP2B6* mediated 8-hydroxylation, with a small contribution from other CYPs (e.g., *CYP2A6, CYP3A*, and *CYP1A2*) [17].

Previously available data from peripheral blood sample studies used efavirenz concentrations as a surrogate endpoint for the metabolic activity of *CYP2B6*. These studies reported an increase in efavirenz plasma concentration, suggesting the decreased metabolic activity of this enzyme [23]. However, a recent study conducted in Kenya [24] has shown that efavirenz concentration may be decreased in part of a HIV-infected population of patients, connected to specific polymorphisms. Efavirenz plasma levels were mostly linked to polymorphisms of *CYP2B6* and although polymorphisms may correlate to gene expression, for *CYP2B6*, there are studies to show that this may not always be the case [18]. When discussing efavirenz as a model for determining *CYP2B6* activity, it is important to mention that efavirenz causes acute inhibition, but demonstrates chronic induction of *CYP2B6* in a genotype-dependent manner, as shown in the study with efavirenz and bupropion in healthy volunteers [25]. Enzyme induction can lead to the increased expression of genes encoding the enzyme in question.

It is important to note that most of the studies that analyze *CYP2B6* activity were conducted using some kind of surrogate endpoint, such as efavirenz plasma concentration, or were performed on tissues such as peripheral blood cells or healthy donor hepatocytes cell culture [22,24,26]. One of the few studies performed on human liver tissue samples in a population of HCV mono-infected patients by Drozdzik et al. showed that there was down-regulation of a couple of CYP enzymes, but the level of *CYP2B6* remained unchanged [27]. Despite the seemingly counterintuitive finding of increased *CYP2B6* expression in HIV/HCV co-infected patients, multiple factors may contribute to this phenomenon. Our study is one of the very few studies on human liver samples in this specific population with chronic inflammation. Through our research using univariant analysis, we evaluated the influence of other known factors, such as liver fibrosis and liver inflammation, as well as smoking, alcohol, and gender, as potential factors affecting gene expression of *CYP2B6* but no correlation was found. Adding to these findings, the lack of correlation of *CYP2B6* with the observed parameters might be a specific quality of the study group, considering our results suggest that *CYP2B6* expression was unaffected by these previously mentioned factors.

### 4.2. Choice of Therapy of HCV Mono-Infected Patients Independently Correlates with Expression of CYP2B6

Upon analyzing gene expression differences for *CYP2B6* in two groups of HCV mono-infected patients, initially, no statistically significant difference was found between the untreated and treated HCV patients. However, when comparing patients treated with DAA and those treated with PEG IFN, a higher expression of *CYP2B6* was observed in the latter group. In addition, when compared with a group of untreated HCV patients, the PEG IFN treated group showed increased *CYP2B6* gene expression as well. Although in vitro studies suggest that certain individuals exhibit decreased expression of *CYP2B6* and increased expression of *CYP3A4* [28,29], limited and conflicting data exist regarding the effects of PEG IFN on *CYP2B6* expression. Our results suggest that the considerable individual variability in *CYP2B6* expression in response to PEG IFN treatment may be a potential contributing factor. Additional factors such as liver damage, age, gender, and *CYP2B6* polymorphisms may also contribute to this variability. However, it is important to note that the small sample size receiving PEG INF in our study is a significant limitation. These findings may well correspond with interferon-free treatment strategies in HCV treatment.

### 4.3. An Increase in CYP3A4 Expression Correlates with the Presence of HIV Infection

*CYP3A4*, as is the case with other metabolic enzymes, has been reported to be downregulated in HIV and HCV infection, although the underlying mechanisms of regulation differ and there is significant inter-individual variation [30]. In the study by Jones et al. [31], using caffeine, dextromethorphan, and midazolam tests, lower *CYP3A4* activity was observed in 17 HIV-infected patients compared to the controls, while Jetter et al. [32] found approximately 50% lower overall CYP3A activity in 30 HIV-infected patients compared to healthy volunteers using midazolam, dextromethorphan, and digoxin in in vivo phenotyping tests. An impact of inflammation on the pharmacokinetics of antiretrovirals mainly metabolized by *CYP3A4* would thus be expected. However, data in the literature on human liver samples are sparse. While it is crucial to understand *CYP3A4* modulation in HIV for safety and efficacy concerns, studies are showing conflicting results. For instance, atazanavir concentrations were found to be lower in HIV-infected patients than in healthy volunteers, and a study analyzing caffeine metabolism found that caffeine metabolism was not altered in HIV-infected patients compared to healthy volunteers but was decreased in AIDS patients, suggesting that the level of inflammation may be an important factor affecting metabolizing enzymes [33,34,35].

We found a positive correlation showing increased *CYP3A4* expression in HIV-infected patients, both mono- and co-infected with HCV. This result can potentially be also interpreted as a correlation to the use of ART, given that all the HIV-infected patients received therapy. Most of our patients were using efavirenz, lamivudine, and dolutegravir (although dolutegravir is only partially metabolized through this enzyme) in different combinations, as the drugs that are mostly metabolized through *CYP3A4* [9,35]. There was no correlation between the specific choice of drug or drug combination with the expression of *CYP3A4*. After observing these data, we are prone to conclude that the *CYP3A4* gene expression increase observed may be activation or induction of a common pathway for this metabolizing enzyme as it was previously shown in studies [9,35]. It is unfortunate and a limitation of this study not to have a control group of untreated HIV-infected patients. This corresponds to the common limitations of most studies, due to ethical considerations. As for other potential factors affecting *CYP3A4*, platelet count was found to be significantly higher in patients with a higher expression of *CYP3A4*. There was no significant correlation (*p* > 0.05) between the changes in the level of expression of *CYP3A4* when compared with other biochemical and clinical parameters analyzed. *CYP3A4* expression was also positively correlated with higher grades of liver inflammation, although no correlation (*p* > 0.05) with the stage of liver fibrosis was found.

### 4.4. Lower ABCB1 Expression Correlates with the Use of Lamivudine

In the present study, a positive correlation was observed between *ABCB1* expression and lamivudine usage in HIV patients. Usage of lamivudine was found to be correlated with lower expression of the *ABCB1* transporter genes, which is in line with the results of previous studies [36]. The regulation of *ABCB1* gene expression is complex and may involve various factors such as viral infection, inflammation, and antiretroviral therapy. Some studies have reported downregulation of *ABCB1* gene expression in HIV and HCV infections [37,38] while others have reported upregulation or no significant change [39,40]. Additionally, inter-individual variations in *ABCB1* gene expression may exist, and pharmacogenomic studies have yielded conflicting data so far [41,42]. The present study holds value as it represents, to the best to our knowledge, one of few studies conducted on human liver tissue samples, specifically in the HIV and HCV populations. Furthermore, the utilization of human liver tissue samples in this specific population provides a unique opportunity to directly examine and elucidate all potential confounding factors affecting the gene expression of metabolic enzymes and transporters in human liver tissue, which cannot be fully captured through in vitro or animal models alone. However, it is important to acknowledge that the study’s relatively small sample size represents a potential limitation. Future studies with larger sample sizes are warranted to confirm and extend these initial findings.

## 5. Conclusions

In conclusion, this study highlights the significant increase in *CYP2B6* gene expression associated with co-infection, a significant association of the presence of HIV infection with the increase in *CYP3A4* mRNA levels, and a trend observed for the downregulation of *ABCB1* in patients using lamivudine. The absence of correlation with liver damage, inflammation, and specific treatment interventions emphasizes the need for additional research to elucidate the complex interplay between gene expression, viral co-infection, liver pathology, and therapeutic responses in this particular population.

## Figures and Tables

**Figure 1 medicina-59-01207-f001:**
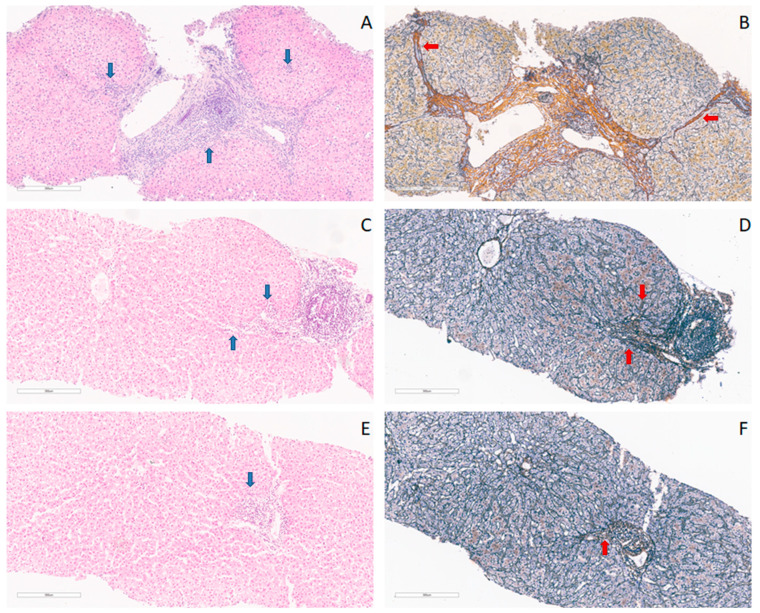
Photomicrographs showing the patohistological analysis. Liver tissue needle biopsy samples and heterogeneous findings, with blue arrows marking necro-inflammatory activity, and red arrows marking reticulin fibers, (**A**) showing moderate necro-inflammatory activity—A2, METAVIR score, from HIV/HCV co-infected patient, hematoxylin–eosin stain, magnification 10×; (**B**) cirrhosis—F4, METAVIR score, from HIV/HCV co-infected patient, reticuline-silver stain, magnification 10×; (**C**) mild necro-inflammatory activity—A1, METAVIR score, from HCV-infected patient, hematoxylin–eosin stain, magnification 10×; (**D**) moderate fibrosis, periportal with few septa—F1, METAVIR score, from HCV-infected patient; reticuline-silver stain, magnification 10×; (**E**) minimal necro-inflammatory activity, hematoxylin–eosin stain—A1, METAVIR score, from HIV-infected patient,, magnification 10×; (**F**) no fibrosis—F0, METAVIR score, from HIV-infected patient, reticuline-silver stain, magnification 10×.

**Table 1 medicina-59-01207-t001:** *CYP2B6* gene expression in groups of HIV, HCV and HIV/HCV co-infected patients.

Groups		*CYP2B6* Expression (Mean ± SD)	*p* Value	Posthoc
HCV+ without HCV therapy		1.711 ± 0.635	0.017	1* vs. 3* *p* = 0.007 2* vs. 3* *p* = 0.030
HIV+		1.813 ± 0.866		
HIV+/HCV+ without HCV therapy		3.673 ± 2.704		
HCV+ without HCV therapy		1.711 ± 0.635	0.252	1* vs. 4* EG IFN *p* = 0.021 1* vs. 4* DAA *p* > 0.05
HCV+ with HCV therapy	Total	2.117 ± 1.081		
	PEG IFN	2.727 ± 1.144	0.028	
	DAA	1.601 ± 0.781		

1*, HCV+ without HCV therapy; 2*, HIV+; 3*, HIV+/HCV+ without HCV therapy; 4*, HCV+ with HCV therapy (total); PEG IFN—pegylated interferon; DAA—direct acting antivirals.

**Table 2 medicina-59-01207-t002:** *CYP3A4* gene expression in a group of HIV, HCV and HIV/HCV co-infected patients.

Groups		*CYP3A4* Expression (Mean ± SD)	*p* Value	Posthoc
HCV+ without HCV therapy		2.377 ± 1.227	0.359	1* vs. 3* *p* > 0.05 1* vs. 2* *p* > 0.05 2* vs. 3* *p* > 0.05
HIV+		3.730 ± 2.696		
HIV+/HCV+ without HCV therapy		4.245 ± 4.799		
HCV+ without HCV therapy		2.377 ± 1.227	0.149	1* vs. 4* PEG IFN *p* > 0.05 1* vs. 4* DAA *p* > 0.05
HCV+ with HCV therapy	Total	1.637 ± 1.252		
	PEG IFN	1.174 ± 0.613	0.300	
	DAA	1.968 ± 1.521		

1*, HCV+ without HCV therapy; 2*, HIV+; 3*, HIV+/HCV+ without HCV therapy; 4*, HCV+ with HCV therapy (total); PEG IFN—pegylated interferon; DAA—direct-acting antivirals.

**Table 3 medicina-59-01207-t003:** *ABCB1* expression in a group of HIV, HCV, and HIV/HCV co-infected patients.

Groups		*ABCB1* Expression (Mean ± SD)	*p* Value	Posthoc
HCV+ without HCV therapy		0.934 ± 0.379	0.116	1* vs. 3* *p* > 0.05 1* vs. 2* *p* > 0.05 2* vs. 3* *p* > 0.05
HIV+		0.903 ± 0.339		
HIV+/HCV+ without HCV therapy		3.652 ± 5.325		
HCV+ without HCV therapy		0.934 ± 0.379	0.466	1* vs. 4* PEG IFN *p* > 0.05 1* vs. 4* DAA *p* > 0.05
HCV+ with HCV therapy	Total	1.060 ± 0.398		
	PEG IFN	0.986 ± 0.588	0.657	
	DAA	1.110 ± 0.267		

1*, HCV+ without, therapy; 2*, HIV+; 3*, HIV+/HCV+ without HCV therapy; 4*, HCV+ with HCV therapy (total); PEG IFN—pegylated interferon; DAA—direct-acting antivirals.

## Data Availability

All data on the patient population in this study are available at request.

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
