# Peer review of "Expression of CYP2B6 Enzyme in Human Liver Tissue of HIV and HCV Patients"

_medicina, 2023, doi:10.3390/medicina59071207_

Round 1

Reviewer 1 Report

In this study, authors collected the information of 54 patients who underwent liver biopsy. They They compared the gene expression of CYP2B6, CYP3A4, and ABCB1 transporter in patients with HIV, HCV, and HIV/HCV co-infection. The methods are well-described and the discussion is comprehensive. However, its significance is limited due to the small number of patients. Some issues should be addressed.

1. More description of HCV, HIV and HCV/HIV infection should be given in Introduction.

2. Some typos were found in the manuscript. Proofreading is suggested.

3. Define abbreviations at first use.

4. Did the authors compare the basic information of these patients? The data was shown in Line 200. But I recommend it would be clear to summarize and compare the basic information as a table.

5. Arrows should be added in Figure 1 to point out the pathological changes. Also, figures must be referred in the main text.

Moderate editing of English language required.

Reviewer 2 Report

In this study, Obradovic et al. analyzed the RNA expression levels in liver biopsies from 54 individuals of CYP2B6, CYP3A4 (two cytochrome P450 group enzymes) and ABCB1 (a transmembrane transporter P-glycoprotein). The participants were separated into three groups: 1) people living with HIV, 2) people co-infected with HIV and HCV, and 3) people infected with HCV only. All individuals living with HIV (irrespective of HCV status) were receiving ART treatment at the time of the study (although not all individuals were receiving the same drugs); none of the co-infected individuals were receiving treatment for the HCV, and the mono-infected with HCV were stratified into 3 groups: those not receiving any HCV treatment, those receiving PEG-IFN, and those receiving DAA. The main findings include an increase of CYP2B6 RNA levels in the group of co-infected people, when compared with the other two groups (HIV or HCV mono-infected). The strengths of the study include a well-defined and characterized cohort. The weaknesses include that the authors did not include the clinical data on the cohorts and that there is no cohort of untreated HIV infected individuals so the findings could be due to the HIV infection or the ART treatment. I found the study of interest, but I think the analysis of these enzymes and transporters are tricky with regard to their interpretation since they can be modulated by so many factors, including alcohol consumption and use of tobacco products. In line with this, the authors cite a number of works with similar results but also some with opposite findings.

I have some comments/suggestions:

o   It would be very informative if the authors could show the relevant clinical data on the 54 study participants (as a supplementary table or the like). Also, what were the reasons for performing a liver biopsy on these subjects? I am just wondering since the reasons were not clearly stated and they may be of relevance to the study. Was the liver biopsy done to confirm HCV infection by any chance?

o   Authors stated that inclusion criteria included serologically confirmed diagnosis of HIV and HCV. Did they run any confirmatory PCR? Probably not so important for the HIV diagnosis but could any individual be potentially cured of their HCV infection -and still positive by antibody tests- at the time of biopsy? 

o   The manuscript would benefit from some editing. There are many typos and errors and some sentences that were difficult to understand. I suggest the authors go through all the text carefully and solve all these issues. Also, I found the discussion especially difficult to follow, so please re-write carefully.

o   Some examples of typos/errors: 

-   Spearmen should read Spearman (line 191), included total should read included a total (line 200), expression in group should read expression in the group (line 230), missing an o in of (line 227), treated with PEG should read treated with PEG IFN (line 251), expression the group should read expression in the (line 319), compared should read comparing (line 333), contrary, recent study should ready on the contrary, a recent (line 374), This should read These (line 412), in study by should read in a study by (line 417), transoprters (line 465), authors mentioned several times along the text “RNK extractions”. I believe RNA extractions is what was meant. Same applies to PEG INF which should read PEG IFN...

o   Some examples of sentences difficult to understand: 

-   Lines 219-221: I found these sentences confusing. Please modify for enhanced clarity. The initial sentences of the other sections looked better than this one. Also, there was no allusion to the inflammation or fibrosis in this section while it was stated in the subsequent ones. Please add some text in that regard for consistency.

-   Lines 235-237: I found these sentences confusing. Please modify for enhanced clarity.

-   Line 386: it was not clear to me what was meant in “or study population was of healthy cell culture”. Please rewrite for enhanced clarity.

-   Lines 393-396: I found these sentences confusing. Please modify for enhanced clarity.

-   Lines 451-453: I found these statements confusing and contradictory. Are they meant to be two separate ideas? If so please add references to the second statement. 

o   Authors mention is some instances “other studies show this or show that” but they do not provide the corresponding cites. Please provide. Ex: lines 456-458.

o   The materials and methods section was repeated. It was added after the introduction and then again after the discussion (although the latter was a partial and slightly different version). Please remove one of the two sections. 

o   Figure 1 doesn’t seem to be called out in the text. Please do so. This figure lacks context and proper explanations; for instance, what type of patient/patients was this from? In the legend it is stated “moderate activity”. I think it would be helpful to the reader if the authors extended and explained what activity they are referring to. Please clarify.

o   Please note that I downloaded a supplementary material file along with the materials for revision but the supplementary file came in a non-standard format that I was unable to open. I contacted the editorial on the issue but I had not received a reply by the time I submitted these comments. No supplementary material was called out in the text so I am assuming this was some sort of technical bug.

The comments on the quality of the English are included in my general comments.

Round 2

Reviewer 1 Report

Authors have revised the manuscript accordingly and the issues have been addressed. The quality of the manuscript has improved. It is suitable for publication after minor editing of language.

Minor editing of English language required.